# Analysis of Exosomal Cargo Provides Accurate Clinical, Histologic and Mutational Information in Non-Small Cell Lung Cancer

**DOI:** 10.3390/cancers14133216

**Published:** 2022-06-30

**Authors:** Elena Duréndez-Sáez, Silvia Calabuig-Fariñas, Susana Torres-Martínez, Andrea Moreno-Manuel, Alejandro Herreros-Pomares, Eva Escorihuela, Marais Mosqueda, Sandra Gallach, Ricardo Guijarro, Eva Serna, Cristian Suárez-Cabrera, Jesús M. Paramio, Ana Blasco, Carlos Camps, Eloisa Jantus-Lewintre

**Affiliations:** 1Molecular Oncology Laboratory, Fundación Investigación Hospital General Universitario de Valencia, 46014 Valencia, Spain; edurendez@incliva.es (E.D.-S.); calabuix_sil@gva.es (S.C.-F.); susana.torres@alu.umh.es (S.T.-M.); moreno_andman@externos.gva.es (A.M.-M.); alherpo@etsiamn.upv.es (A.H.-P.); escorihuela_eva@gva.es (E.E.); mosqueda_mar@gva.es (M.M.); gallach_sangar@gva.es (S.G.); 2TRIAL Mixed Unit, Centro Investigación Príncipe Felipe—Fundación Investigación Hospital General Universitario de Valencia, 46014 Valencia, Spain; blasco_ana@gva.es; 3Centro de Investigación Biomédica en Red Cáncer, CIBERONC, 28029 Madrid, Spain; guijarro_ricjor@gva.es (R.G.); cristian.suarez@ciemat.es (C.S.-C.); jesusm.paramio@ciemat.es (J.M.P.); 4Department of Pathology, Universitat de València, 46010 Valencia, Spain; 5Department of Biotechnology, Universitat Politècnica de València, 46022 Valencia, Spain; 6Department of Surgery, Universitat de València, 46010 Valencia, Spain; 7Department of Thoracic Surgery, Hospital General Universitario de Valencia, 46014 Valencia, Spain; 8Freshage Research Group, Department of Physiology, Universitat de València, 46010 Valencia, Spain; eva.serna@uv.es; 9Centro de Investigación Biomédica en Red Fragilidad y Envejecimiento Saludable CIBERFES, Fundación Investigación Hospital Clínico Universitario/INCLIVA, 46010 Valencia, Spain; 10Biomedical Research Institute i+12, Hospital Universitario “12 de Octubre”, 28040 Madrid, Spain; 11Molecular Oncology Unit, CIEMAT, 28045 Madrid, Spain; 12Department of Medical Oncology, Hospital General Universitario de Valencia, 46014 Valencia, Spain; 13Department of Medicine, Universitat de València, 46010 Valencia, Spain; 14Joint Unit: Nanomedicine, Centro Investigación Príncipe Felipe—Universitat Politècnica de Valencia, 46022 Valencia, Spain

**Keywords:** non-small cell lung cancer, liquid biopsy, exosomes, extracellular vesicles, cell cultures, adenocarcinoma, squamous cell carcinoma, biomarker, tumorspheres

## Abstract

**Simple Summary:**

Non-small cell lung cancer (NSCLC) is the second most commonly diagnosed cancer and the leading cause of cancer-related death worldwide. Clinical decision-making depends on the histological classification; however, tissue biopsy is frequently not technically feasible due to tumor location or limited tissue samples. Therefore, we propose to find clinical, molecular and histological biomarkers using a minimally invasive approach based on the analysis of the cargo of the blood extracellular vesicles. Exosomes are membranous vesicles present in several biological fluids, which carry biological information to distant tissues, regulating several tumor processes. This study aims to analyze NSCLC exosome cargo for search biomarkers that could improve clinical management. This report demonstrates the possibility of implementing exosomes to detect molecular alterations and as a source of biomarkers to differentiate NSCLC histology, allowing for a new approach in precision oncology.

**Abstract:**

Lung cancer is a malignant disease with high mortality and poor prognosis, frequently diagnosed at advanced stages. Nowadays, immense progress in treatment has been achieved. However, the present scenario continues to be critical, and a full comprehension of tumor progression mechanisms is required, with exosomes being potentially relevant players. Exosomes are membranous vesicles that contain biological information, which can be transported cell-to-cell and modulate relevant processes in the hallmarks of cancer. The present research aims to characterize the exosomes’ cargo and study their role in NSCLC to identify biomarkers. We analyzed exosomes secreted by primary cultures and cell lines, grown in monolayer and tumorsphere formations. Exosomal DNA content showed molecular alterations, whereas RNA high-throughput analysis resulted in a pattern of differentially expressed genes depending on histology. The most significant differences were found in XAGE1B, CABYR, NKX2-1, SEPP1, CAPRIN1, and RIOK3 genes when samples from two independent cohorts of resected NSCLC patients were analyzed. We identified and validated biomarkers for adenocarcinoma and squamous cell carcinoma. Our results could represent a relevant contribution concerning exosomes in clinical practice, allowing for the identification of biomarkers that provide information regarding tumor features, prognosis and clinical behavior of the disease.

## 1. Introduction

Extracellular vesicles (EVs), which have a wide range of sizes and different cellular origins, are cell-released membrane particles regulating intercellular communication by transporting functional molecules. These membrane-bound vesicular structures contain substantial amounts of biologically active information acquired from their parental cells, which can be transported to other cells or organs in homeostasis or disease conditions [1]. EVs are classified into different groups of microvesicles, based on their morphological features and content.

In particular, exosomes are a subset of small EVs (~40 to 160 nm) secreted by different cells. The biogenesis of exosomes involves their origin in endosomes, and the subsequent interactions of exosomes with other intracellular vesicles and organelles determine their final content. The diverse contents, inside and on the surface, may include nucleic acids, proteins, lipids, amino acids, and metabolites, reflecting the cell type of origin [2]. In diverse diseases, exosome-mediated molecular mechanisms and their detection in biological fluids potentially offer a means to devise novel and more effective treatment strategies [3]. The potential of EVs to be a source of biomarkers for diagnosis, prognosis, and surveillance processes has been proposed. Accumulating evidence suggests that cancer-derived EVs can be detected in different body fluids, such as blood, urine, and saliva, among others. Several members of the tetraspanin protein family, including CD9, CD63, and CD81, are highly enriched in the lipid bilayer surfaces of exosomes. Interestingly, transmembrane integrins and other proteins displayed on the surfaces of exosomes determine the target cell type and contribute to organotrophic displacement [4]. In addition, exosomes contain abundant and diverse nucleic acids such as DNA, messenger RNAs (mRNAs), and microRNAs (miRNAs), the last being the most abundant cargo and consequently the focus of most exosome biomarker research and functional studies. Other small RNAs, such as long non-coding RNAs (lncRNAs), long intergenic RNAs (lincRNAs), and circular RNAs, are also present [5,6,7]. Cancer-derived exosomes are implicated in various cancer processes, including pre-metastatic niche formation and metastasis, development of the tumor microenvironment, angiogenesis, evasion of immune surveillance, and acquisition of aggressive phenotypes. All the above reasons encourage their study as a relevant source of diagnostic and prognostic non-invasive biomarkers in cancer [8,9,10,11].

3D in vitro models have been proposed as new approaches to allow for the study of different mechanisms such as drug resistance, tumor progression, self-renewal, and metastasis, and might be a closer approximation to in vivo processes [12]. Specifically, the tumorspheres model has been widely accepted for determining self-renewal capability and for the identification of new therapeutic molecules capable of annihilating populations of cells. In addition, the 3D structure of this model accurately represents how cancer cells are arranged and grow inside the tumor, compared to monolayer cell cultures [13]. In this regard, tumorsphere-derived exosomes have demonstrated great ability to provide molecular information and therapeutic benefits in lung cancer [14], which is a leading cause of death and the most common malignancy worldwide [15]. Indeed, despite advances in the understanding of cancer biology and the development of new therapeutic agents, most cases of lung cancer are diagnosed at an advanced stage (non-resectable). As a result, five-year overall survival rate for non-small cell lung cancer (NSCLC) is less than 20%. Unfortunately, even at early stages of NSCLC, approximately 30% of patients suffer recurrent disease, and many patients die after surgical treatment [16].

Newly identified molecular profiles and druggable genetic alterations in lung cancer gave rise to a modification of the World Health Organization (WHO) classification in 2015. Most lung cancers are detected at an advanced stage, with small biopsy or cytology specimens being diagnosed following WHO categorization. Within the main histological subtypes of NSCLC, lung adenocarcinoma (LUAD) was defined as carcinoma with an acinar/tubular structure or mucin production, positive for TTF-1 (NKX2-1) immunochemistry (IHC) and/or Napsin A, whereas lung squamous cell carcinoma (LUSC) was defined as carcinoma with keratinization or intercellular bridges with CK5/6, p40 and p63 positive. When possible, the differential diagnosis between LUAD and LUSC is useful for selecting a therapy based on targetable driver genetic alterations [17]. However, in up to 25% of conventional tumor biopsies, it is not possible to determine the complete molecular profile of the tumor, since tissue samples often do not meet the quality control criteria and/or have insufficient tumor material to carry out the necessary determinations. In addition, these types of biopsies provide limited information on the dynamics of tumor evolution and the different cell populations that comprise the tumor. There is an additional problem, since biopsies can rarely be repeated sequentially due to tumor location or size, and carry a risk of procedure-related complications, especially in patients with advanced stages.

The latest therapeutic advances focus on the concept of cancer as a heterogeneous disease [18]. The proportion of cancer patients with tumors harboring potentially targetable genomic abnormalities at diagnosis or during progression continues to increase to date [19,20,21]. Hence, we need other tools to complement the absence of tissue biopsies. Extensive research has focused on the identification of driver mutations (*EGFR*, *ALK*, *MET*…) and their inhibition, as well as on the implementation of new treatments based on immunotherapy [22,23,24,25]. Although those treatments have improved survival rates among patients with NSCLC, the prognosis remains poor, and new therapeutic approaches are required. In this scenario, liquid biopsy arises as a non-invasive diagnostic platform that can establish tumor molecular profiles at the beginning of treatment and during tumor evolution, acting as a source of biomarkers that provides us with a global vision of the heterogeneity of the tumors [26]. Liquid biopsy constitutes a great diversity of biological fluids and offers multiple elements for study, among which exosomes stand out [27,28]. The future direction of precision oncology requires molecular research to improve treatment options [29]. Therefore, exploring novel biomarkers from EVs would be helpful for selecting patients and improving prognosis in different NSCLC stages. The present study is a proof-of-concept focused on the use of exosomes as a minimally invasive platform to search for biomarkers in NCSLC. For this purpose, we initially used in vitro models under different growth conditions, and then confirmed the results in silico and validated them in a cohort of NSCLC patients. There is still a need to identify exosome-based biomarkers, not only for histological diagnosis, but also for patient prognosis stratification and mutational status monitoring throughout treatment with targeted therapies. The accomplishment of these objectives will enhance the efficient management of advanced NSCLC.

## 2. Materials and Methods

### 2.1. Patients and Tissue Samples

This study included 186 patients from the General University Hospital of Valencia who underwent surgery between 2004 and 2016. Lung tumor specimens were obtained at the time of surgery and fitted the following eligibility criteria: candidate for surgical resection, non-pretreated, over 18 years, non-pregnant, stage I–IIIA (according to the American Joint Committee on Cancer staging manual) with a histological diagnosis of NSCLC, and with informed consent obtained. Some patient tumor samples were immediately processed for primary culture establishment and the rest were preserved in RNAlater (Applied Biosystems, Waltham, MA, USA) at −80 °C until further analyses.

Demographic and clinicopathological characteristics of all patients were collected. Follow-up was performed according to the institutional standard for resected NSCLC. The study was conducted in accordance with the Declaration of Helsinki, and the ethical review board of the General University Hospital of Valencia approved the protocol (3 July 2017). All patients signed informed consent forms for sample acquisition and research purposes.

### 2.2. Establishment of Primary Cell Cultures

Surgical tumor specimens from patients were successfully established, being able to grow tumor cells as monolayers and tumorspheres. For this study, four primary patient-derived lung cancer cultures were employed (Appendix A). Tumor dissociation was previously described by our group [13]. Tumor profiling of each patient-derived culture was determined by next-generation sequencing (NGS) using Oncomine Focus Assay (Thermofisher Scientific, Waltham, MA, USA) and Ion GeneStudio S5 System (Thermofisher Scientific, Waltham, MA, USA) to obtain complete tumor profiling of each patient.

### 2.3. Commercial NSCLC Cell Lines

Thirteen human NSCLC cell lines (A549, NCI-H1395, NCI-H1650, NCI-H1975, NCI-H2228, NCI-H358, NCI-H460, HCC827, NCI-H520, NCI-H1703, LUDLU-1, SK-MES-1, PC9, and SW900) were used for in vitro experiments. LUAD cell lines and the LUSC SW900 cell line were purchased from American Type Culture Collection (ATCC, Manassas, VA, USA), whereas the rest of the LUSC cell lines were kindly provided by Dr. J. Carretero (University of Valencia, Spain) (Appendix A). All cell cultures (primary and commercial) were tested for mycoplasma before the start of the experiments.

### 2.4. Tumor Cell Culture Growth Conditions

Tumor cells grown in monolayer (2D cultures) were cultured in DMEM-F12 (primary cultures) or RPMI-1640 (commercial cells lines) containing 10% FBS, 200 µg/mL penicillin/streptomycin, 2 mM L-glutamine (for DMEM-F12), and 0.001% non-essential amino acids (for RPMI-1640). For the formation of tumorspheres (3D cultures), cells grown in monolayer were trypsinized using 0.05% trypsin-EDTA at 80% confluence. Cells were seeded at low density in ultra-low-attachment flasks (Corning, NY, USA) with serum-free medium (DMEM-F12/RPMI-1640) supplemented with 0.4% BSA, 50 µg/mL EGF, 20 µg/mL bFGF, 5 µg/mL, ITS PREMIX, 2% B27, and 200 µg/mL penicillin/streptomycin (Gibco™, Grand Island, NY, USA).

### 2.5. In Silico Dataset Validation

In silico analysis was performed using different lung cancer datasets from The Cancer Genome Atlas (TCGA) consortium [30,31]. RNA-seq data and clinical information were downloaded from the ICGC Data Portal, https://dcc.icgc.org/releaes/current/projects/LUAD-US accessed on 10 June 2020, and https://dcc.icgc.org/releases/current/projects/LUSC-US accessed on 10 June 2020. The Limma package from Bioconductor was used to obtain normalized RNA-seq data. Linear fit models for the genes were obtained before constructing the different contrast matrixes. Given the linear models, empirical Bayes statistics were computed for differential expression analysis.

### 2.6. Isolation of Exosomes from Cell Cultures

To isolate tumor-derived exosomes obtained from cell cultures, cells were grown in T175 cm^2^ flasks until 70–80% confluence for 72 h in 30 mL of FBS-depleted media (in the case of monolayer cultures). After 72 h, the supernatant was differentially centrifugated at 500× *g* for 5 min, and then at 3000× *g* for 15 min to eliminate cell detritus. Afterwards, the supernatant was filtered through a 0.2 µm filter (Corning, NY, USA) and ultracentrifuged at 110,000× *g* for 90 min (CP-NX, P50AT2 Rotor; Hitachi, Japan). A second ultracentrifugation was performed to wash the first obtained pellet; exosomes were then resuspended in 30 mL of PBS. All centrifugations were performed at 4 °C. Finally, exosomes were resuspended in a small volume (30–60 µL) of filtered PBS and stored at −80 °C until the corresponding analysis.

### 2.7. Negative-Staining and Transmission Electron Microscopy (TEM) of Exosomes

An amount of 6 μL of exosomes resuspended in PBS was placed onto Formvar carbon-coated grids and contrasted with 2% uranyl acetate. Sample preparations were visualized using a FEI Tecnai G2 Spirit transmission electron microscope (FEI Europe, Eindhoven, The Netherlands). Imaging was performed using a Gatan UltraScan US1000 CCD camera and data was analyzed with Digital Micrograph 1.8 (Gatan, Ametek, Berwyn, IL, USA).

### 2.8. Nanoparticle Tracking Analysis (NTA)

Size distribution and concentration of isolated exosomes were measured using a NanoSight NS300 instrument (Malvern Instruments, Ltd., Malvern, UK). The instrument was calibrated using silica microspheres of different diameters prior to sample analysis. To perform the measurements, samples were uniformly diluted 10-to 100-fold in a 0.1 µm-filtered PBS solution in order to reduce the number of particles in the field of view to fewer than 100 per frame. This technique combines laser light scattering microscopy with a charge-coupled device (CCD) camera, which enables the visualization and recording of nanoparticles in solution. The nanoparticle tracking analysis (NTA) software is then able to identify and track individual nanoparticles moving under Brownian motion and relates the movement to the particle size using the following formula derived from the Stokes–Einstein Equation (1) [32]:(1)$$\bar{\left(x,y\right)2}=\frac{2k_{B}T}{3r_{h}\pi \eta }$$
where **k****_B_** is the Boltzmann constant and $\bar{\left(x,y\right)2}$ is the mean-squared speed of a particle at a temperature **T**, in a medium of viscosity **η**, with a hydrodynamic radius of **r****_h_**. Readings were taken five times over 60 s at 10 frames per second at room temperature. Approximately 3 × 10^8^ particles/mL of sample were examined to assess the size distribution and concentration. The resulting data were studied using nanoparticle tracking analysis software v3.1 with camera level set to 10 and detection threshold to 5 (Malvern Instruments Ltd., Amesbury, UK, http://www.malvern.com/ accessed on 9 July 2020).

### 2.9. Immunoblotting

Cell culture and isolated exosome pellets were lysed using a lysis buffer composed of 50 mM Tris-HCl pH 7.5, 150 mM NaCl, 0.02% NaN3, 0.1% sodium dodecyl sulfate (SDS), 1% NP40, 0.5% sodium deoxycholate, 2 mg/mL leupeptin, 2 mg/mL aprotinin, 1 mM phenylmethylsulfonyl fluoride (PMSF), and protease inhibitor cocktail (Roche, Basel, Switzerland). The Bradford assay (Sigma-Aldrich, St. Louis, MO, USA) was employed to quantify the total protein concentration; 25 μg of total protein were separated on 10% SDS–polyacrylamide gel and electro-transferred to a 0.45 μm polyvinylidene difluoride (PVDF) membrane (MilliporeSigma, Burlington, MA, USA). The membrane was then blocked with 5% skim milk for 1 h and immunoblotted overnight at 4 °C with CD9 anti-rabbit (ab223052, Abcam, Cambridge, UK), TSG101 anti-mouse (ab125011), Calnexin anti-rabbit (ab75801), and β-actin anti-mouse (A5441, Sigma-Aldrich, St. Louis, MO, USA) antibodies. Afterwards, membranes were incubated with anti-IgG (whole molecule)-Peroxidase secondary antibodies (Sigma-Aldrich, St. Louis, MO, USA; Thermo Fisher Scientific, Waltham, MA, USA) for 1 h at room temperature (Appendix A). For chemiluminescent detection, the high-sensitivity Amersham ECL Select™ detection reagent (GE Healthcare, Chicago, IL, USA) was used.

### 2.10. Flow Cytometry Analysis

Isolated exosomes were incubated for 1 h with human CD81-PE and human CD63-APC antibodies (Miltenyi Biotech, Bergisch Gladbach, Germany) (Appendix A) in PBS. The negative control for background fluorescence was performed using the same antibodies incubated in PBS. After incubation, exosomes were acquired using a FC500 MPL Flow Cytometer and CytExpert v2.3 software (Beckman-Coulter, Inc., Brea, CA, USA). Finally, samples with positive staining for the antibodies of interest were treated with Triton 0.01% for 15 min at RT to lyse the exosomes’ lipid bilayers. This was performed to confirm that previously obtained signal mostly disappeared, and could serve as a control to dismiss positivity due to cell debris.

### 2.11. RNA Isolation and Integrity

RNA from cell culture pellets and tumor frozen tissue samples was extracted using the standard TRI Reagent^®^ RNA Isolation Reagent, (Sigma-Aldrich, St. Louis, MO, USA) method. Exosomal total RNA derived from cell cultures was isolated using the Total RNA Purification Kit (Norgen Biotek, Thorold, ON, Canada). RNA integrity and concentration were assessed with the Agilent 2100 Bioanalyzer (Agilent, Santa Clara, CA, USA), using RNA 6000 Nano and Pico Kit (Agilent, Santa Clara, CA, USA).

### 2.12. Exosomal Mutational Status in Established NSCLC Primary Cultures

For *EGFR*, *KRAS* and *BRAF* mutation analysis, the BEAMing digital PCR technique (Beads, Emulsions, Amplification, and Magnetics) (Sysmex Inostics, Inc., Baltimore, MD, USA) was used. DNA was extracted from exosomes using the QIAamp DNA Micro kit (Qiagen, Hilden, Germany) according to the manufacturer’s instructions. DNA concentration was quantified using the Qubit dsDNA high sensitivity assay, according to the manufacturer’s protocol (Thermo Fisher Scientific, Waltham, MA, USA). Starting from the exosomal RNA previously extracted, *ALK* gene rearrangements present in the cell culture samples were determined using the *ALK* Gene Fusions and *ROS1* Gene Fusions Detection Kit (Amoy Diagnostics, Xiamen, China); mRNA was transcribed to cDNA at 42 °C for 1 h and the gene fusion was readily detected using quantitative real time PCR (RT-qPCR), according to the manufacturer’s protocol.

### 2.13. Whole mRNA Expression Profiling

Total RNA was isolated from H1650, H1975, H2228, SW900, FIS 301 and FIS 343 cell culture-derived exosomes grown in 2D or 3D conditions (analyzing two biological replicates for each sample), using TRI Reagent^®^ RNA Isolation Reagent (Sigma-Aldrich, St. Louis, MO, USA). Two biological replicates were performed. An amount of 13 ng of exosomal RNA was then amplified, labelled, and hybridized using the Clariom^TM^ D Assay, human (Thermofisher, Waltham, MA, USA), following the manufacturer’s instructions. Briefly, the Affymetrix GeneChip WT Pico Kit was used for cDNA preparation and biotin labelling. Arrays were incubated for 16 h in an Affymetrix GeneChip 645 hybridization oven at 45 °C with rotation at 60 rpm, and subsequently scanned using an Affymetrix^®^ GeneChip Scanner 3000 7G at 570 nm wavelength excitation. Input files were normalized to eliminate systematic sources of variation that were not differences in expression (efficiency in color marking, amount of RNA, and spatial effects of the chip, among others), using the robust multiple-array average (RMA) algorithm in Affymetrix Expression Console and Transcriptome Analysis Console 4.0 software. CEL files were used to analyze significant changes in gene expression profiles and were statistically filtered using Partek Genomic Suite 6.6 software (Partek Inc., Chesterfield, MO, USA). Afterwards, a one-way ANOVA was performed and statistically significant genes were identified using a model analysis of variance of Fold Discovery Rate (FDR). Data were deposited in the Gene Expression Omnibus (GEO)—NCBI database: GSE198238.

Pathway enrichment analysis was performed on the differentially expressed genes (DEGs) obtained by histology using Pathway Studio. Lists of the pathological processes assigned to each of these gene sets were trimmed at approximately the 100 lowest *p*-values (*p* < 0.01), with four being the minimum number of overlapped DEGs for each process. Data were rendered in a bubble plot using the Ggplot2 package of RStudio (RStudio, Inc., Boston, MA, USA).

### 2.14. DEGs Validation Using RT-qPCR

Reverse transcription–quantitative real time PCR was performed to validate the relative expression of the most significant differentially expressed genes in transcriptome microarrays and reference genes (Appendix A). This step was performed using a LightCycler^®^480 II system (Roche, Basel, Switzerland). Reverse transcription reactions were performed using 500 ng (cells and tissue samples) and 150ng (exosomes samples) of total RNA, random hexanucleotides, and the High-Capacity cDNA (complementary DNA) Reverse Transcription Kit (Applied Biosystems, Waltham, MA, USA), following the manufacturer’s instructions. RT-qPCR was performed with assays based on hydrolysis probes using 1 μL of cDNA, 2.5 μL of TaqMan Gene Expression Master Mix and 0.25 μL of TaqMan Gene Expression Assay (Applied Biosystems, Waltham, MA, USA) to a 5 μL final reaction volume. To calculate the efficiency, random-primed qPCR Human Reference cDNA (Takara Bio, Mountain View, CA, USA) was used. *ACTB*, *GUSB*, and *CDKN1B* were selected as endogenous controls using GeNorm software (https://genorm.cmgg.be/ accessed on 9 July 2015) for tissue analysis, whereas *ACTB* and *GAPDH* were selected as endogenous controls for exosome samples. Relative gene expression levels were expressed as the ratio of target gene expression to the geometric mean of the endogenous gene expressions, according to the Pfaffl formula [33].

### 2.15. Immunofluorescence Analysis

Cells were fixed in 4% paraformaldehyde in PBS at room temperature for 15 min, washed with PBS, permeabilized with 0.4% Triton X-100 in PBS for 10 min, and washed again with PBS. Permeabilized cells were blocked with PBS containing 1% BSA for 1 h, and subsequently incubated with XAGE1 anti-goat [1:100] (ab27477, Abcam, Cambridge, UK) and CABYR anti-rabbit [1:100] (ab 243417, Abcam, Cambridge, UK) antibodies in blocking buffer overnight at 4 °C. Thereafter, cells were washed with PBS and incubated with Alexa-labelled IgG secondary antibodies (Appendix A) for 1 h. Slides were incubated with 4′,6-diamidino-2-phenylindole for 3 min, mounted with Fluoromount Aqueous Mounting Medium (Sigma-Aldrich, St. Louis, MO, USA), and analyzed using a Leica confocal microscope (Leica Microsystems, Buffalo Grove, IL, USA).

### 2.16. Immunohistochemical Analysis

Sections 4 µm thick were obtained from the most representative formalin-fixed, paraffin-embedded (FFPE) blocks of each NSCLC tumor for analysis. Immunohistochemical staining was performed using the standard technique of antigen retrieval and development of avidin–biotin–peroxidase complexes (ABC). Briefly, 4-µm tissue sections were deparaffinized in xylene and mounted on Poly-L-lysine-coated slides. All slides were subjected to a heat-based antigen retrieval method using DAKO Target Retrieval Solution (Agilent, Santa Clara, CA, USA), containing 10 mM citrate buffer (pH 6), and a water bath (95–99 °C) for approximately 20 min before immunostaining. The primary antibody to TTF-1, DAKO clone 8G7G3/1 (Agilent, Santa Clara, CA, USA), was used at the manufacturer’s recommended dilution (1:200). Slides were counterstained with hematoxylin. Sections of TTF-1-positive LUAD were used as positive controls. The primary antibody was replaced with diaminobenzidine (3,3′-diaminobenzidine) solution for the negative controls. TTF-1 nuclear staining was graded as negative (<5%), weak positive + (5–49%), or strong positive ++ (>50%) based on the percentage of tumor nuclei with unequivocal staining.

### 2.17. Statistical Analysis

Non-parametric Mann–Whitney U and Kruskal–Wallis tests were used to compare continuous variables, Spearman’s rank test was used to assess correlations between continuous variables, while the associations between discrete variables were evaluated by the χ^2^ test. Survival analyses were performed using univariate Cox regression analysis and Kaplan–Meier (log-rank) tests, with clinical pathological variables and gene expression levels dichotomized using the median as a cut-off value. A probability of 95% (*p* < 0.05) was considered statistically significant for all analyses. Statistical analyses were performed using the Statistical Package for the Social Sciences (SPSS, Chicago, IL, USA) version 15.0 and GraphPad Prism version 6.0.

## 3. Results

### 3.1. Characterization of Exosomes Derived from NSCLC Cell Cultures

A total of 4 primary cell cultures from NSCLC patients and 13 lung cancer commercial cell lines, grown in 2D monolayer and 3D tumorsphere conditions, were used to characterize tumor-derived exosomes (Appendix A). To determine the efficiency of the methodology employed for the isolation of these microvesicles, several techniques for their characterization were used. First, the mean diameter of isolated microvesicles was determined using NTA with a concentration of 1 × 10^8^–1 × 10^9^ particles/mL. Microvesicles were 110 ± 156 nm (Figure 1a), which is consistent with the typical diameter of exosomes. Moreover, these microvesicles showed the typical morphology of an exosome, rounded and cup-shaped, according to TEM (Figure 1b).

In addition, exosome-specific markers TSG101, CD9, CD63 and CD81 were evaluated in tumor-derived exosomes through different methodologies. On the one hand, immunoblot analyses revealed that exosomes derived from cell cultures co-expressed CD9 and TSG101 (tumor susceptibility gene 101 protein), whereas they were negative for Calnexin, which was used as a negative control (endoplasmic reticulum marker) (Figure 1c) (Appendix A). On the other hand, the positive expression of exosomal surface markers CD63 and CD81 (tetraspanin superfamily of activation-linked cell surface antigens) was also confirmed by flow cytometry (Figure 1d).

Exosomes secreted by 2D monolayer cultures and 3D tumorspheres were compared based on different parameters. Regarding morphology, no differences were found between microvesicles derived from these two cell cultures according to TEM imaging (Figure 1b). However, exosomes secreted by 3D cultures were larger in size compared to 2D-derived exosomes (median size 123.65 nm 2D vs. 132.5 nm 3D (*p* = 0.022)). In addition, some surface markers, such as CD63 and CD81, showed higher expression in exosomes secreted from 3D cultures (Figure 1d).

### 3.2. Mutational Status of Exosomes Derived from Cell Cultures

We wondered if the presence of *EGFR, KRAS*, or *BRAF* mutations, as well as *ALK* rearrangements, could be detected in exosome cargo. We performed dPCR using isolated exosomes from tumor cells grown in 2D conditions. Mutation status correlated 100% (10 of 10) between culture cells and tumor-derived exosomes (Figure 2 and Appendix A).

### 3.3. Differential Expression Profiles of Tumor Cell Culture-Derived Exosomes

To investigate the expression patterns of tumor cell-derived exosomes, we conducted a transcriptomic study using whole genome expression microarrays. The results obtained revealed the presence of a large number of mRNAs, small RNAs and long non-coding RNAs in exosomes. Supervised analysis of the microarray dataset allowed for the detection of different expression profiles among samples. In the first step, principal component analysis (PCA) allowed us to obtain a global vision of the distribution of the samples among the different variables. Interestingly, PCA evidenced grouping of samples between LUAD and LUSC histologies (Figure 3a). To gain biological insights from the differentially expressed probe sets according to histology, we performed further hierarchical clustering analyses on the microarray dataset using an arbitrary fold change (FC) cutoff of >1 and significance *p*-value of ≤0.01. We found 551 genes overexpressed in LUAD exosomes, with a maximum FC of 4.38, and 803 overexpressed in LUSC, with a maximum FC of 4.72 (Figure 3b). We then restricted the FC cut-off to ≥1.5, observing 46 probes overexpressed in LUAD exosomes and 293 probes overexpressed in LUSC.

In the next step, we focused on highly expressed genes in LUAD (*XAGE1B* and *SEPP1*) and LUSC (*CAPRIN1, RIOK3* and *CABYR*) derived exosomes for further validation. We also validated TTF-1 (NKX2-1), since it is an established LUAD marker and routinely used for the determination of this histological subtype in NSCLC (among other types of tumors) (Table 1).

Afterwards, pathway enrichment analysis was performed for the differentially expressed genes observed in the microarray histology comparison to identify the main processes in which they were involved. A total of 13 pathological process were significantly enriched (*p* ≤ 0.03) (Figure 4). All these pathways are related to different hallmarks of cancer (Appendix A).

### 3.4. Biomarker Validation in Tumor Cell Culture-Derived Exosomes

To confirm the findings obtained from analysis of the expression microarrays, these candidate genes were examined in a larger number of exosome samples from NSCLC cell cultures using quantitative RT-PCR (RT-qPCR).

Using this technique, it was confirmed that *XAGE1B* presented significantly (*p* = 0.01) higher expression in LUAD-secreted exosomes (N = 11, Figure 5a), while there was no expression of this gene in the LUSC group (N = 6), except for the cell line SKMES-1, which exhibited low expression values (Figure 5b). RT-qPCR results for *CABYR* were also consistent with the transcriptomic findings. Expression of this gene was found in LUSC cell-derived exosomes, while no expression was detected in LUAD cell-secreted exosomes (*p* < 0.001) (Figure 5c,d).

However, when comparing the expression of TTF-1, SEPP1 (Appendix A) RIOK3 and CAPRIN1 (Appendix A) between LUAD and LUSC histologies, no significant differences in their expression were found (*p* > 0.05).

According to the cell culture models, transcriptomic differences were also observed in exosomes derived from 2D versus 3D cultures. PCA and hierarchical cluster analysis showed two groups of differentially expressed genes according to the growth model used in cell cultures from which the exosomes were isolated (Appendix A). Interestingly, exosomes derived from LUAD cell cultures grown in 3D conditions (enriched in CSCs) showed a significant higher expression of XAGE1B (*p* = 0.013) in comparison to 2D-monolayer (Appendix A). No differences were observed in the expression of the other genes analysed for these two models (data not shown).

Given the remarkable differential expression of XAGE1B and CABYR at mRNA level in exosomes from LUAD and LUSC, respectively, we assessed their expression at protein level through immunofluorescence (IF) analysis in two primary cell cultures (FIS 471 and FIS 301). As shown in Figure 6, higher protein expression of XAGE1B was found in the cytosol of FIS 471 (LUAD), while the FIS 301 (LUSC) exhibited greater expression for CABYR, in accordance with the transcriptomics and RT-qPCR results.

### 3.5. In Silico Validation of Exosomal Biomarkers in NSCLC

In order to continue inquiring into the relevance of these previously described genes, we analyzed them in an independent cohort of NSCLC patients from the TCGA database (The Cancer Genome Atlas Program). The clinicopathological characteristics of the patients included in this study are summarized in Table 2.

Patients with post-surgical complications were excluded from the survival analysis, and only those patients who had at least one month of follow-up were included (N = 661). Mann–Whitney U tests indicated a relevant higher expression of *XAGE1B, SEPP1* and *TTF-1 (NKX2-1)* (*p* < 0.001) in the group of LUAD patients (N = 328), compared to the LUSC group (Figure 7a–c).

Conversely, the expression of *CABYR* and *RIOK3* showed significantly higher values (*p* < 0.001) (Figure 7d,e), and *CAPRIN1* showed a minor difference (*p* = 0.037) in the group of patients with LUSC (N = 316) compared with the LUAD cohort (Figure 7f). No other significant associations were found between gene expression and survival or other clinicopathological variables (data not shown).

### 3.6. Validation of Exosomal Biomarkers in a Resected NSCLC Cohort

To further validate the potential value of *XAGE1B, SEPP1, TTF-1, CABYR, RIOK3* and *CAPRIN1* as biomarkers, an independent cohort of patients with resected LUAD and LUSC tumors from HGUV was used. Clinicopathological characteristics of the resected patient cohort are summarized in Table 3. The median follow-up of the patients was 31.93 months [range: 1–161.7].

In this cohort of resected NSCLC patients, analysis of TTF-1 expression was routinely performed via IHC at the Anatomical Pathology Department of the HGUV. All LUAD patients included in the cohort were selected based on this marker and excluding other LUSC markers already established for this diagnosis in the department. Afterwards, we analyzed the expression of the other markers selected in the study by RT-qPCR, upon extraction of genetic material from the same pieces of tumor tissue used for the formation of FFPE blocks for IHC analyses. Among the LUAD group markers, only *XAGE1B* remained significant for this histology (*p* < 0.001), showing a clear identification of the relative expression cutoff values to differentiate between LUAD and LUSC (Figure 8a). However, no statistically significant differences were observed regarding *SEPP1* relative expression of *SEPP1* between both groups (data not shown).

By contrast, *CABYR* and *RIOK3* were also more expressed in LUSC patients (*p* = 0.003 and *p* = 0.022, respectively) (Figure 8b,c). *CAPRIN1* hardly showed any expression differences between both histologies (non-significant, data not shown).

Given the results obtained in this cohort, we selected only those markers that presented very notable differences in relative expression cutoff values between both histological groups (*XAGE1B* and *CABYR*). In consequence, the next step in our work was to assess if the expression of these genes could predict the main histological subtypes. In addition, we determined their potential role in distinguishing between lung tumor tissues and healthy tissue. Interestingly, Wilcoxon analysis of the resected NSCLC patients with paired samples revealed a higher presence of *XAGE1B* (*p* = 0.003) and *CABYR* (*p* < 0.001) in lung tumor tissue versus normal adjacent tissue (NAT) (Figure 9).

Finally, we assessed the role of these genes as potential prognostic factors in the HGUV NSCLC patient cohort. Kaplan–Meier analysis showed a significant association of XAGE1B with patient prognosis. Patients with higher expression levels of *XAGE1B* (>median) had shorter relapse-free survival (RFS) (21.13 vs. NR months, *p* = 0.022) and overall survival (OS) (49.63 vs. not reached (NR) months, *p* = 0.013) in the LUAD group (N = 74) (Figure 10). In contrast, *CABYR* was not significantly correlated with prognosis factors (data not shown).

## 4. Discussion

Despite recent advances in the treatment of NSCLC, prognosis remains very poor due to delay in the detection of the disease [16]. Numerous patients are diagnosed in advanced stages of the disease and tumor sampling is often unavailable for histopathological diagnosis. Therefore, there is a fundamental necessity to find new approaches to enable the accurate diagnosis of lung cancer.

Currently, available lung cancer treatments present certain shortcomings; thus, new cost-effective methods are urgently needed in clinical practice. Furthermore, although clinical decision-making depends on the histological and molecular classification, tissue biopsy is frequently not technically feasible due to tumor location or scant tissue samples. In this sense, we proposed to find clinical, molecular and histological biomarkers using different strategies, such as exosomes. Exosomes are a promising tool to assess biomarkers in the scenario of NSCLC, since they contain genetic material with valuable biological information. Due to their endocytic origin, exosome cargo reflects the composition of the parental tumor cells, making it a potential substitute for tumor biopsies [34].

The present paper is a proof-of-concept study that reveals the feasibility of using exosome-based biomarkers in the clinical management of NSCLC. Several studies have corroborated the usefulness of lung cancer-derived exosomes as representative elements of the cell of origin in many features [35,36]. In this work, we propose exosomes as a tool for the study of NSCLC, as they present several advantages compared to other, better-implemented elements of liquid biopsy such as cell-free DNA (cfDNA) or Circulating Tumor Cells (CTCs). For instance, exosomes present a lipid bilayer, which gives them stability in the bloodstream or during bulk storage, preserving their content against degradation. In addition, their presence in biological fluids is greater compared to CTCs and their genetic material provides more information than cfDNA fragments alone [37,38].

At present, exploring biomarkers in NSCLC cell lines is still in the pre-clinical phase. Although research studies based on NSCLC cell lines have revealed promising results for their application as biomarkers, they still present limitations that should be overcome before clinical translation [39,40]. To obtain an in vitro model that represents the complex physiological behavior of a tumor it is necessary to draw on human primary cells isolated directly from tissues in order to retain the histological, biological and functional characteristics of their origin tissue. In the present study, we optimized the isolation and multiple analysis of exosomes using NSCLC cell cultures. Notably, we employed primary patient-derived cell cultures from our hospital, which were a suitable platform as described by Kodack et al., Zhang et al. and Kim et al. [41,42,43]. Cell cultures were used to obtain exosomes since they are a stable, easy-to-use and fast study platform, which allows for the obtention of large amounts of exosomes that are sufficient to carry out all the subsequent analyses. Non-adherent or ultra-low-attachment three-dimensional (3D) cultures, also called sphere formation assays, have been widely used to assess the potential stemness of cancer cells and their three-dimensional structure [44]. For this purpose, our study also aimed to determine valuable biomarkers in exosomes secreted by different populations of NSCLC tumor cells using two different growth models (2D vs. 3D)**.** Previous studies by our group analyzed and determined the potential of spheroids as a platform to understand the complexity of lung tumors via in vitro models [13,45,46]. Even so, the limitations of these models must also be taken into account. Cell lines likely represent a homogeneous subpopulation of the original tumor, due to selective survival pressures present in culture conditions that do not exist in the original microenvironment. The lack of interactions with stromal, immune, and inflammatory cells within a three-dimensional (3D) environment also limits the translational potential of cell line studies [47].

In this work, we corroborated through different methodologies the correct means of obtaining extracellular vesicles secreted by both cell culture models (2D and 3D). The isolation of the EVs was carried out by ultracentrifugation. To date, there are still no well-established methods for this procedure; however, it seemed to us the most appropriate methodology due to its good reproducibility and the absence of chemical reagents that can potentially interfere with downstream analysis of EVs [48]. By analyzing the size, morphology and concentration of our samples, as well as identifying specific determined surface markers (TSG101, CD9, CD63 and CD81), we were able to observe that the EV samples obtained were enriched in tumor-derived exosomes, comparable with other previously published studies [49,50].

We analyzed the presence of specific mutations in extracellular vesicles secreted by lung primary cultures as well as cell lines and were able to detect the same genetic alterations previously determined in the tumor cells of origin. These results confirm the potential of NSCLC-derived exosomes as a tool for determining the mutational status of the most relevant genes in the clinical management of this pathology, as supported by previous studies [51,52]. Our next objective was to identify new potential markers in NSCLC. Given that cancer-derived exosomes carry a wide variety of RNAs, we conducted a transcriptome-level expression profiling assay with >540,000 transcripts. This adds value to the current knowledge of exosomes because the majority of studies are focused on miRNAs and lncRNAs [53,54,55]. In our study, we found differentially expressed genes on exosomes of LUAD and LUSC in 2D and 3D cell models. After validating these preliminary results in 2D and 3D models from cell cultures, we next aimed to assess this differential gene expression in histological subgroups of an NSCLC patient cohort. For this purpose, we performed in silico validation using The Cancer Genome Atlas (TCGA) database (N = 661). The present study confirmed three DEGs that were overexpressed in patients with LUAD compared with those with LUSC, and three DEGs that were downregulated in patients with LUAD compared with those with LUSC. However, there are currently limited available studies on effective molecular diagnostic markers of LUAD and LUSC [56,57]. For example, Luccheta et al. used a robust statistical framework with the integration of diverse bioinformatics tools to analyze NGS data from more than 1000 patients from LUAD vs. LUSC [58]. Their results found upregulation of O-glycosylation mucin genes in LUAD and genes related to immune response in LUSC. In our study using TCGA, the number of samples employed was minor due to the use of tumor tissue samples, as well as adjacent normal tissue, to obtain accurate dysregulated genes in LUAD and LUSC. To further validate the potential value of *XAGE1B, SEPP1, TTF-1, CABYR, RIOK3* and *CAPRIN1* as biomarkers, we analyzed them using another independent cohort of patients. This patient cohort consisted of paired blood and tumor tissue samples, as well as adjacent normal tissue, and presented more than 10 years of follow-up. Indeed, we previously validated different biomarkers using this patient cohort [59,60,61,62]. In the present study, we validated *XAGE1B* as dominant in LUAD whereas *CABYR* was validated as a LUSC biomarker.

Both *XAGE1B* and *CABYR* are tumor-specific antigens of the Cancer Testis Antigens (CTA), which have attracted research attention as potential mediators of cancer cell recognition. CTAs are expressed in a variety of cancers, including lung cancer, while in normal tissues their expression is restricted to immune-privileged sites, such as the testis and placenta [63]. For this reason, they are considered ideal targets for cancer treatment due to their highly immunogenic and restricted expression in germ cells and malignancies [64,65].

Our study incorporated *XAGE1B* as a poor prognostic biomarker in the resected LUAD cohort of patients. Further LUAD subgroup analysis of RFS and OS indicated that high *XAGE1B* expression was significantly correlated with a worse prognosis. In accordance with these results, CTAs were previously associated with poor prognosis and advanced stages of NSCLC [66]. Although *XAGE1* was previously described in NSCLC adenocarcinoma [67,68], our study revealed the presence and high expression of *XAGE1B* in EVs secreted by parental tumor cells. The expression profile of this biomarker, detected in elements such as exosomes, opens an alternative door for us to be able to determine the histological type of patients in cases where it is not possible using a conventional biopsy. Interestingly, the expression of *XAGE1B* was significantly higher in exosomes from 3D vs. 2D cultures. In other lines of our work, we have observed that EVs derived from 3D cultures showed significantly different profiles, in terms of secretion dynamics and signaling molecular content, compared to those derived from 2D cultures. Many of them modulate several pathways involved in tumor progression and/or related to the population of CSCs. Previous studies corroborate that EVs actively participate in cell-to-cell interactions by shutting down cellular components. There was evidence that CSC-derived EVs carry the stemness markers of parent cells, reprograming non-CSCs to acquire stem-like properties and leading to enhanced tumorigenicity [69,70,71].

The value of exosomes as prognostic biomarkers is not yet widely studied. One meta-analysis indicated that the expression level of exosomes was closely associated with the OS and DFS of patients with lung cancer, proposing that lung cancer exosomes are associated with poor prognosis [72]. The presence of this and other markers in liquid biopsy elements, such as exosomes, provides a great advantage in the management of NSCLC. The next step is to validate these results in an independent cohort of NSCLC patients in order to implement exosomes as a future tool in clinical practice.

## 5. Conclusions

This study highlights the capacity of exosomes to be used as a tool for assessing diagnostic and prognostic markers in NSCLC. The abovementioned qualities of exosomes make them a comprehensive and suitable platform to analyze DNA, RNA, and proteins from unique samples at unique times. Exosome cargo can reflect the molecular signatures of the tumor cells from which they were secreted. Additionally, we identified differentially expressed genes in the cargo of LUAD-derived exosomes vs. LUSC, which were *XAGE1B* and *CABYR*, respectively. We also validated in silico and with our cohort these genes, which could be used to classify LUAD and LUSC in NSCLC cancer patient samples. Furthermore, we described *XAGE1B* as a prognosis gene in independent cohorts of NSCLC patients.

Consequently, this tumor-derived exosome mediated tumor process appears to be a promising new approach and is a valid tool to incorporate into translational applications. In conclusion, exosomes are a potential tool for finding new diagnostic and prognostic biomarkers for NSCLC, although more studies with larger numbers of patients are still needed to support the present results. In the near future, the use of exosomes in minimally invasive samples may enhance precision medicine and increase the knowledge of lung cancer.

## Figures and Tables

**Figure 1 cancers-14-03216-f001:**
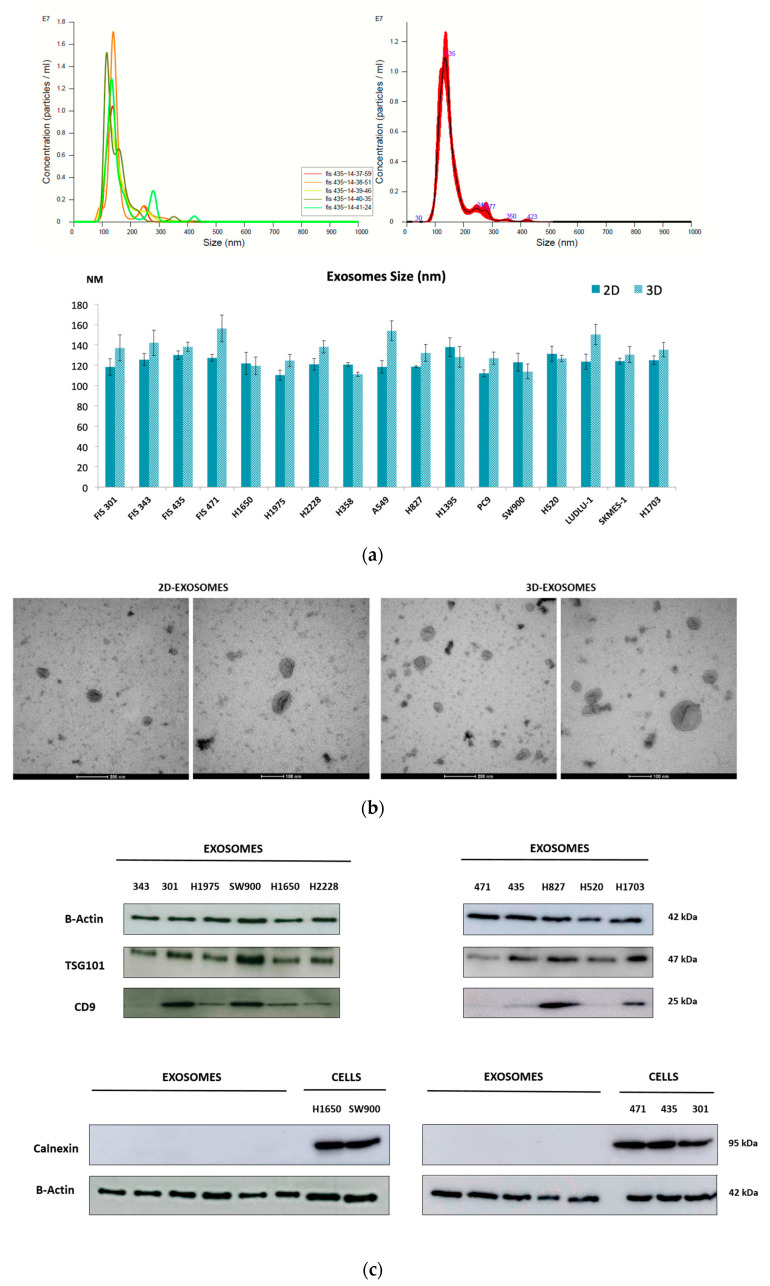
Characterization and quantification of isolated exosomes from tumor cell cultures. (**a**) Representation of the concentration and size distribution obtained using a NanoSight NS300 instrument after analysis of sample FIS 435-2D. Readings were taken five times over 60 s at 10 frames per second at room temperature. An average of all recordings was calculated and is represented in red (top panel). Average size (nm) and standard deviation (SD) were calculated for 2D and 3D exosome samples (bottom panel). (**b**) Representative transmission electron microscopic images of H520-derived exosomes (2D and 3D cell cultures) from NSCLC tumor cells. (**c**) Immunoblotting analysis for exosomal surface markers TSG101 and CD9. Calnexin was used as a negative control for exosome samples (using cell lines H1650 and SW900 as controls and FIS 471, 435 and 301 as primary cultures). β-Actin was used to assess equal protein loading. (**d**) Flow cytometry analysis of surface markers CD63 and CD81 in H520-derived exosomes isolated from 2D and 3D cell cultures.

**Figure 2 cancers-14-03216-f002:**
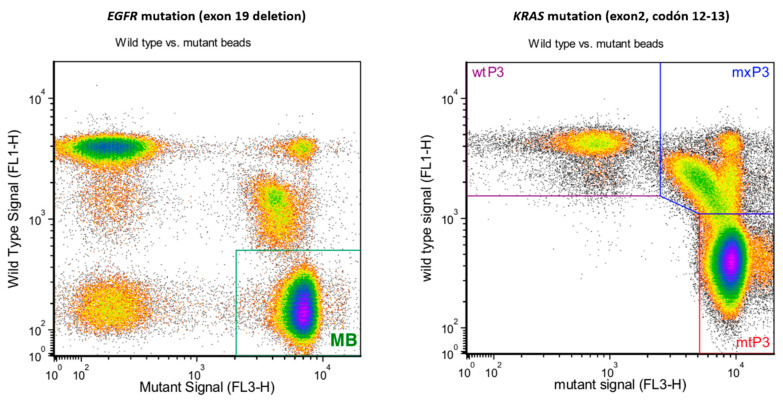
Mutational status determination of *EGFR* and *KRAS* genes using BEAMing technology. A mutant allelic fraction (MAF) of 55.33% for the EGFR exon 19 deletion was detected in exosomal DNA from the H1650 cell line (**left** panel), whereas a mutant fraction of 79.12% was present in SW900 exosomal DNA for a KRAS mutation (exon 2, codon 12–13) (**right** panel).

**Figure 3 cancers-14-03216-f003:**
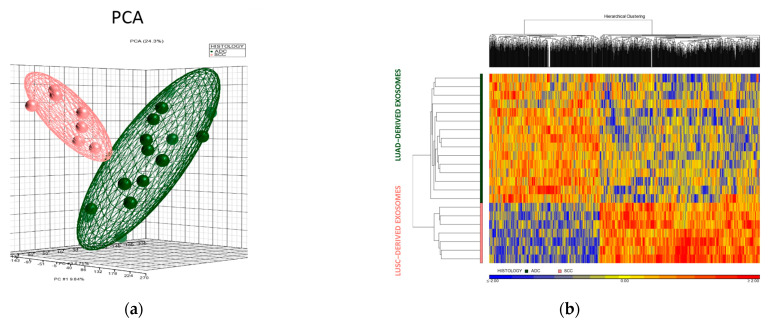
Transcriptomic microarray analysis of cargo of tumor cell culture-derived exosomes. (**a**) PCA plot of H1650, H1975, H2228, SW900, FIS 301 and FIS 343 exosome samples under 2D and 3D conditions (using two biological replicates for each sample) distributed according to histology. (**b**) Hierarchical cluster analysis of differentially expressed probes between LUAD and LUSC. Red color represents overexpression and blue represents underexpression. Rows correspond to the exosome samples analyzed while columns represent the probes detected throughout the samples.

**Figure 4 cancers-14-03216-f004:**
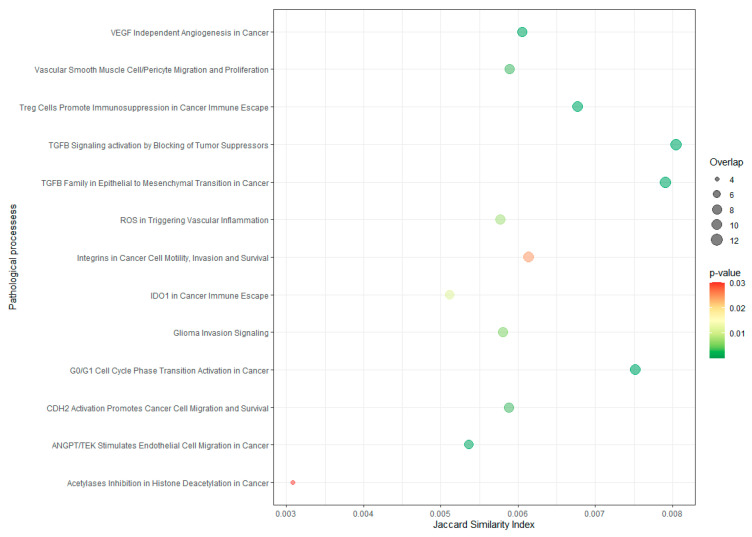
Pathological process enrichment in differentially expressed genes seen in tumor culture-derived exosomes. *Y* axis indicates the name of the pathological process and *X* axis indicates the Jaccard similarity index (JSI). The bubble size indicates the number of overlapped genes for each process. The color bar indicates the *p*-value, representing in red the highest value (0.03), whereas values lower than 0.01 are represented in green.

**Figure 5 cancers-14-03216-f005:**
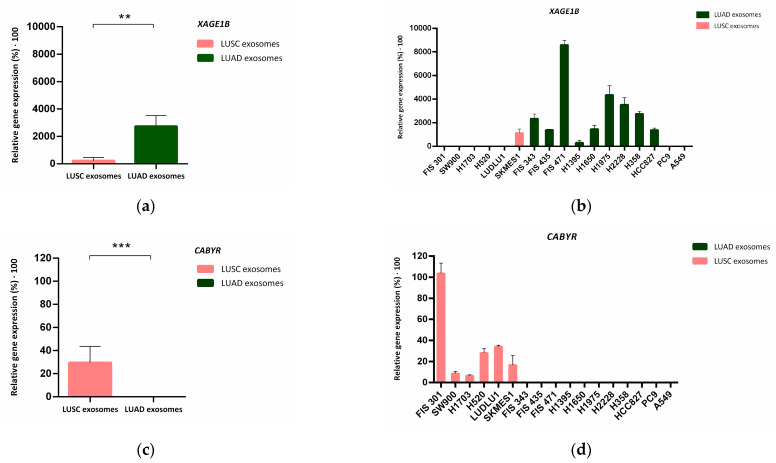
Validation of *XAGE1B* and *CABYR* expression in tumor-derived exosomes from 2D cell cultures. (**a**–**c**) Median of relative gene expression of XAGE1B and CABYR measured via RT-qPCR in both histological groups. Error bars represent SD. Dark green bars correspond to LUAD-derived exosomes while salmon bars correspond to exosomes derived from LUSC samples. (**b**–**d**) Mean with SD of de relative gene expression of *XAGE1B* and *CABYR* to reference genes *ACTB* and *GAPDH* analysed in the complete group of cell cultures-derived exosomes. Significance values were ** *p* ≤ 0.01, and *** *p* ≤ 0.001.

**Figure 6 cancers-14-03216-f006:**
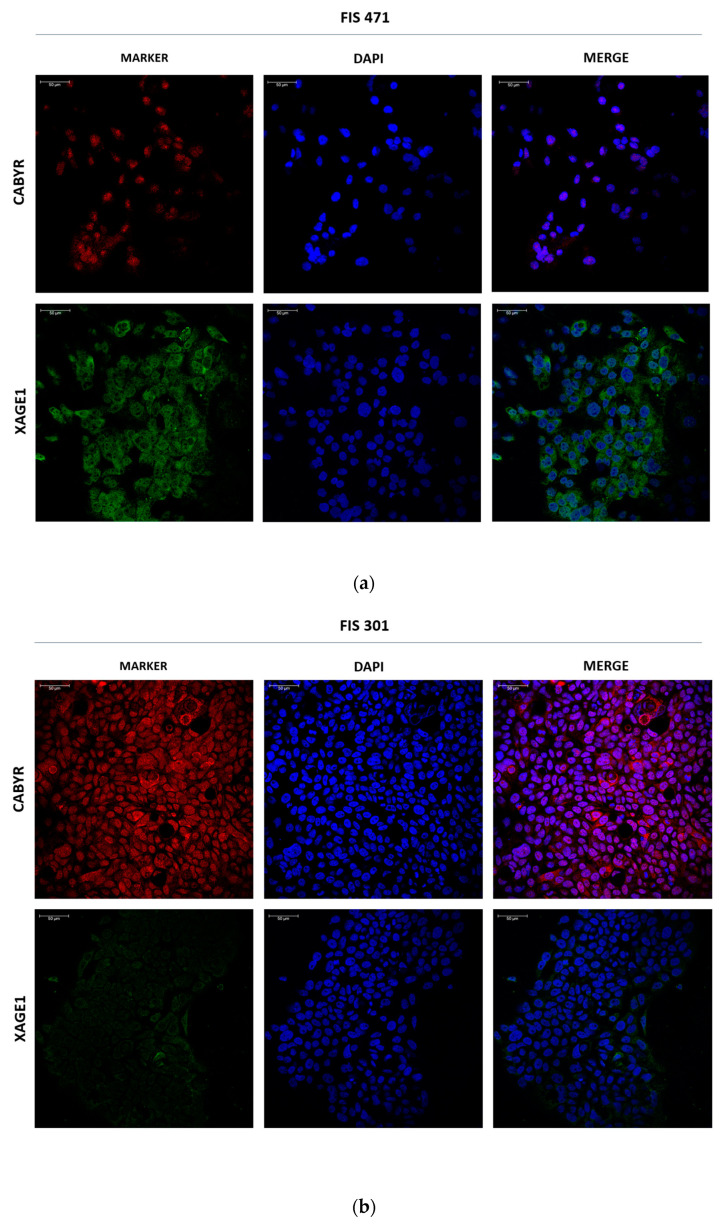
Immunofluorescent staining of CABYR and XAGE1 in primary cultures. (**a**) Representative images of CABYR (red) and XAGE1B (green) in adherent-cultured cells from FIS 471 (LUAD). (**b**) Representative images of CABYR (red) and XAGE1B (green) in adherent-cultured cells from FIS 301 (LUSC). Cell nuclei are stained with DAPI (blue). Scale bar represents 50 μm.

**Figure 7 cancers-14-03216-f007:**
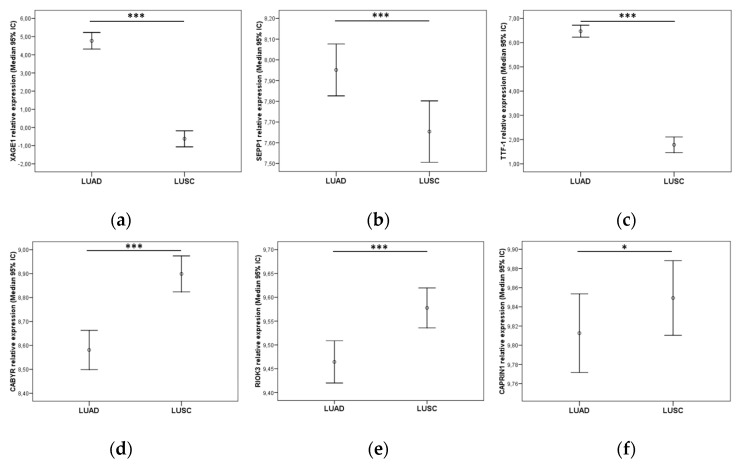
Relative expression of (**a**) *XAGE1*, (**b***) SEPP1*, (**c**) *TTF-1* in the TCGA LUAD cohort and (**d**) *CABYR*, (**e**) *RIOK3*, (**f**) *CAPRIN1* in the LUSC cohort. Graphics represent median with 95% CI (confidence interval). *p*-values were obtained using Mann–Whitney U-tests. Significance values were * *p* < 0.05 and *** *p* ≤ 0.001.

**Figure 8 cancers-14-03216-f008:**
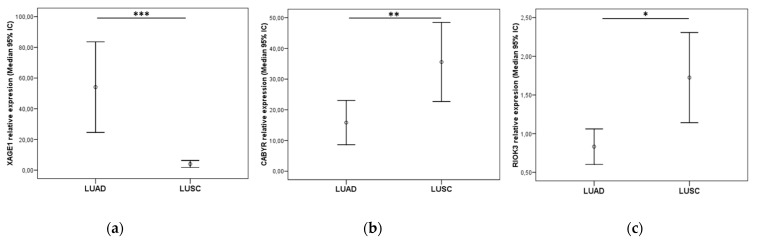
(**a**) *XAGE1* relative expression in the resected NSCLC LUAD cohort and (**b**) *CABYR* and (**c**) *RIOK3* in the LUSC cohort. Graphics represent median with 95% CI (confidence interval). *p*-values were obtained using Mann–Whitney U-test. Significance values were * *p* < 0.05, ** *p* ≤ 0.01, and *** *p* ≤ 0.001. NS: Non-significant.

**Figure 9 cancers-14-03216-f009:**
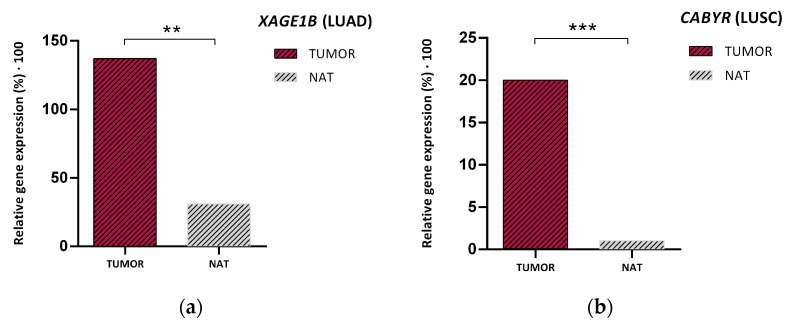
Median relative expression of (**a**) *XAGE1B* and (**b**) *CABYR* in tumor tissue from the HGUV NSCLC resected cohort vs. its expression in NAT (normal adjacent tissue). *p*-values were obtained with ** *p* ≤ 0.01, and *** *p* ≤ 0.001.

**Figure 10 cancers-14-03216-f010:**
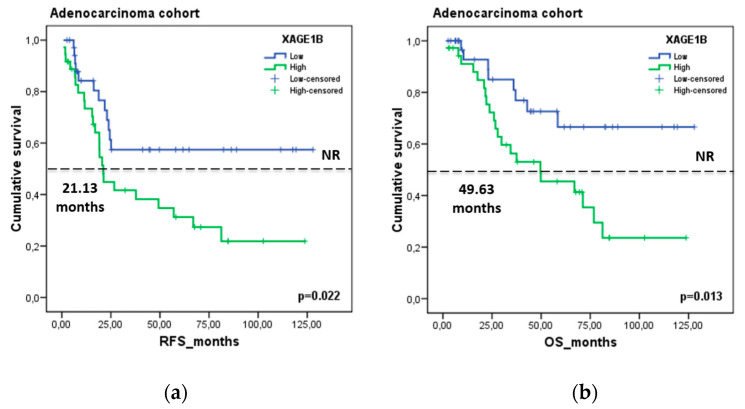
Prognostic value of *XAGE1B* in the HGUV NSCLC cohort. Kaplan–Meier plot for (**a**) RFS and (**b**) OS according to gene expression relative to reference genes (*ACTB, GUSB, CDKN2A*) in the LUAD group. Blue lines represent patients with higher levels of *XAGE1B* expression, whereas green lines represent patients with lower expression levels. Cutoff values correspond to the median relative expression. *p*-values were calculated using the Kaplan–Meier test.

**Table 1 cancers-14-03216-t001:** Main DEGs for each histology screened using Clariom D assays.

Gene Symbol	RefSeq	*p*-Value	Fold Change	Description
*XAGE1B*	NM_001097604	0.0002	4.38139	LUAD up vs. LUSC
*SEPP1*	NM_001085486	0.0042	1.75403	LUAD up vs. LUSC
*NKX2-1 (TTF-1)*	NM_001079668	0.0022	1.27234	LUAD up vs. LUSC
*CAPRIN1* *RIOK3*	NM_005898 NM_003831	8.94 × 10^5^ 0.0001	2.47485 2.56426	LUSC up vs. LUAD LUSC up vs. LUAD
*CABYR*	NM_001308231	2.04 × 10^5^	4.72413	LUSC up vs. LUAD

**Table 2 cancers-14-03216-t002:** Clinicopathological characteristics of the patients included in the in silico cohort (TCGA database).

Characteristics	Total	%
	(N = 661)	
Age at surgery	(median, range)	
	66 (38–88)	
Gender		
Male	395	59.76
Female	266	40.24
Smoking status		
Current	165	24.96
Former	382	57.79
Never	114	17.25
Stage		
IA	152	22.99
IB	223	33.74
IIA	63	9.53
IIB	116	17.55
IIIA	107	16.19
Histology		
Adenocarcinoma	345	52.19
Squamous cell carcinoma	316	47.81
Exitus		
Yes	261	39.49
No	400	60.51

**Table 3 cancers-14-03216-t003:** Clinicopathological characteristics of the patients included in the HGUV NSCLC cohort.

Characteristics	Total	%
	(N = 186)	
Age at surgery	(Median, range)	
	65 (26–85)	
Gender		
Male	158	84.95
Female	28	15.05
Smoking status		
Current	91	48.65
Former	74	39.78
Never	21	11.29
Stage		
I	96	51.61
II	55	29.57
IIIA	35	18.82
Histology		
Adenocarcinoma	79	42.48
Squamous cell carcinoma	90	48.38
Others	17	9.14
Relapse		
Yes	85	45.7
No	101	54.3
Exitus		
Yes	91	48.92
No	95	51.08

## Data Availability

The data presented in this study are available in this article and attached Appendix A.

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
