# Peer review of "Analysis of Exosomal Cargo Provides Accurate Clinical, Histologic and Mutational Information in Non-Small Cell Lung Cancer"

_cancers, 2022, doi:10.3390/cancers14133216_

Round 1

Reviewer 1 Report

The review presents the results of a study aimed at examining their role of exosomes in NSCLC to identify biomarkers. The authors analyzed exosomes secreted by primary cultures and cell lines grown in monolayer and with the formation of tumor spheres. The most significant differences were found in the XAGE1B, CABYR, NKX2-1, SEPP1, CAPRIN1, and RIOK3 genes when analyzing samples from two independent cohorts of resected NSCLC patients. The authors' results may represent an important contribution of exosomes to clinical practice, allowing the identification of biomarkers that provide information about tumor features, prognosis, and clinical course of the disease. I liked the article, the design of the study was well planned, the methods used are described in detail, the results are logically and consistently presented, the conclusions correspond to the data obtained. A small remark concerns figure 1a, very small, poorly perceived.

Reviewer 2 Report

This paper reports the analysis of exosomes cargo in non-small cell lung cancer. This is an interesting attempt for cancer research. However, the present quality of the manuscript is insufficient for publication, please consider the following suggestions that might help to improve the manuscript.

Specific comments:

1. Minor revision of the English language should be performed throughout the manuscript.

2. Figs 1d and  4 were missing/disappear.

3. How was the sample size determined? This information should be explained in the Materials and Methods section.

4. What are the inclusion and exclusion criteria in the study?

5. Future explanation on the choice of 4 NSCLC primary cultures in Table S1.

6. If ethics committee approval, please provide the relevant protocol number in the manuscript.

7. The authors are requested to discuss more about the differential expression genes between LUAD and LUSU in 2D and 3D cell models.

8. Discussion and Conclusions: The authors should rewrite this section in comprehensive and suitable style.

9. There are many reports on the study of exosomes and their role of biomarker in NSCLC so far. Compared with these studies, what does the author think is the innovation of this manuscript?

Round 2

Reviewer 2 Report

The author replied carefully to all reviewer comments. The manuscript can be accepted for publication.